# Influence of Drought and Heat Stress on Mineral Content, Antioxidant Activity and Bioactive Compound Accumulation in Four African *Amaranthus* Species

**DOI:** 10.3390/plants12040953

**Published:** 2023-02-20

**Authors:** Mmbulaheni Happiness Netshimbupfe, Jacques Berner, Frank Van Der Kooy, Olakunle Oladimeji, Chrisna Gouws

**Affiliations:** 1Unit for Environmental Science and Management, North-West University (Potchefstroom Campus), Potchefstroom 2520, South Africa; 2Centre of Excellence for Pharmaceutical Sciences (Pharmacen™), North-West University, Potchefstroom 2520, South Africa

**Keywords:** *Amaranthus* spp., antioxidant activity, drought stress, flavonoids, heat stress, mineral content, phenolics

## Abstract

Drought and heat stress is known to influence the accumulation of mineral content, antioxidant activity, phenolics, flavonoids and other bioactive compounds in many tolerant leafy vegetables. *Amaranthus* plants can tolerate adverse weather conditions, especially drought and heat. Therefore, evaluating the influence of drought and heat stress on commercially and medically important crop species like *Amaranthus* is important to grow the crop for optimal nutritional and medicinal properties. This study investigated the influence of drought and heat stress and a combination of both on the accumulation of phenolic and flavonoid compounds and the antioxidant capacity of African *Amaranthus caudatus*, *A. hypochondriacus*, *A. cruentus* and *A. spinosus*. Phenolic and flavonoid compounds were extracted with methanol and aqueous solvents and were quantified using liquid chromatography with tandem mass spectrometry (LC-MS/MS). Caffeic acid was the main phenolic compound identified in aqueous extracts of *A. caudatus* and *A. hypochondriacus*. Rutin was the most abundant flavonoid compound in all the *Amaranthus* species tested, with the highest concentration found in *A. caudatus*. The results suggest a strong positive, but species and compound-specific effect of drought and heat stress on bioactive compounds accumulation. We concluded that heat stress at 40 °C under well-watered conditions and combined drought and heat stress (at 30 °C and 35 °C) appeared to induce the accumulation of caffeic acid and rutin. Hence, cultivation of these species in semi-arid and arid areas is feasible.

## 1. Introduction

A number of bioactive compounds in plants belong to a class of secondary metabolites with a polyphenolic structure which stressed plants produce as a defensive mechanism [1,2,3]). The impact of drought and heat stress on bioactive compound accumulation, composition and biological activity are still poorly understood in many plant species. The type and concentration of bioactive compounds are determined by the plant’s genetic make-up, developmental stage and the type of stressors they are exposed to [4,5,6].

The impact of global warming makes frequent drought and heat stress a major concern for agricultural production [7,8]. Several studies have indicated that drought and heat stress can enhance the quantitative and qualitative levels of nutrients and bioactive compounds such as flavonoids, resulting in increased antioxidant activity in many tolerant leafy vegetables, including *Amaranthus* [4,9,10,11,12,13]. The chemical diversification of a natural product mixture in response to drought and heat stress can therefore result in an increased level of biological activity [14,15,16,17]. Drought and heat stress influence the accumulation and concentration of highly potent antioxidants like caffeic acid and rutin in many tolerant leafy vegetables [5,18,19,20,21]. These molecules has demonstrated multiple pharmacological activities such as antioxidant, cardioprotective, cytoprotective, vasoprotective, anticarcinogenic and neuroprotective activities [22,23,24]. However, solvents and method of extraction when preparing plant extracts that involve increased temperature, pressure and time can negatively affect the composition, concentration, and biological activity of bioactive compounds [25,26].

Drought and heat stress have differential effects on the nutritional value and bioactive compound accumulation, composition, and biological activity of crop species [4,11,12,13,16,17]. The synergistic influence of these stressors on the nutritional value and production of bioactive compounds in *Amaranthus* are still poorly understood, and little emphasis has been placed on understanding how drought and heat stress specifically affect the biological activity of African *Amaranthus* spp. *Amaranthus* is an underutilised tolerant crop with significant levels of dietary minerals, vitamins and bioactive compounds with potential health benefits. Under current climatic conditions *Amaranthus* offer under developed countries an opportunity to contribute to agricultural sustainability, food security and fighting poverty and malnutrition [27,28,29,30].

Although the influence of drought and heat stress on mineral content and bioactive compound accumulation has been comprehensively studied across different plants [4,10,11,12,13,14,15,25,26], the comparative analysis of the interactive effect at different intensities of drought and heat stress on the mineral content and bioactive compound accumulation across temperature regimes have not been investigated. Therefore, this article explores the interactive effect of drought and heat stress and a combination of both on the mineral content and bioactive compound accumulation for selected African *Amaranthus* species. Furthermore, this study seeks to demonstrate that methanol is a better solvent to extract flavonoid compounds, while an aqueous solution is better for organic acids extraction.

## 2. Results

### 2.1. Effect of Drought and Heat Stress on Mineral Content

Drought and heat stress influenced the accumulation of mineral contents in all tested *Amaranthus* species. K and Ca contents were significantly (*p* ≤ 0.05) higher under control conditions compared to plants treated with 35 °C and 40 °C temperature regimes (Figure 1A–H). In contrast, the Na and NO_3_− contents were substantially increased in *A. cruentus* (26.6% and 53.3% at 30 °C) and *A. spinosus* (120.0% and 43.8% at 30 °C) under well-watered conditions (Figure 1C,D).

All species had higher values of K, Ca, Na and NO_3_- contents at 30 °C and 35 °C, but these values varied significantly (*p* ≤ 0.05) under well-watered and drought-induced conditions. Both species showed higher sensitivity to heat stress (at 40 °C), as the accumulation of K, Ca and NO_3_- decreased significantly (*p* ≤ 0.05) under well-watered conditions when compared to their respective controls (Figure 1A–D). Conversely, the accumulation of these minerals under combined drought and heat stress decreased (Figure 1E–H). The accumulation of K and NO_3_- in *A. cruentus* decreased by 43.6% and 34.3%, respectively, whereas Ca increased by 51.2% under similar conditions (Figure 1G).

Further, it was noted that increasing heat stress under well-watered conditions significantly (*p* ≤ 0.05) increased Na content in *A. caudatus*, *A. hypochondriacus* and *A. spinosus*. In contrast, the Ca content was significantly (*p* ≤ 0.05) lower for *A. caudatus*, *A. hypochondriacus*, *A. cruentus* and *A. spinosus* under well-watered conditions in all temperature regimes compared to drought induced conditions (Figure 1A–H). The highest increase was observed for *A. cruentus* at 40 °C under drought-induced conditions (Figure 1G).

### 2.2. Effect of Drought and Heat Stress on Total Antioxidant Capacity (TAC), Phenolic (TPC) and Flavonoid (TFC) Content Accumulation

The influence of drought and heat stress on the accumulation of TAC, TPC and TFC of *A. caudatus*, *A. hypochondriacus*, *A. cruentus* and *A. spinosus* is presented in Figure 2A–D, Figure 3A–D and Figure 4A–D. TAC (DPPH) was significantly (*p* ≤ 0.05) higher in *A. hypochondriacus* and *A. cruentus* for both methanol and aqueous extracts under control conditions compared to plants treated with 30 °C, 35 °C and 40 °C temperature regimes. It was noted that there was a significant (*p* ≤ 0.05) increase of TAC in *A. caudatus*, *A. hypochondriacus* and *A. spinosus* as the temperature increased under well-watered and drought-induced conditions. TAC was substantially increased in the aqueous extracts of *A. caudatus* compared to methanol extracts. It increased by 48.6% for the control, and 56.6% for 30 °C, 120.6% for 35 °C and 51.4% for 40 °C temperature regimes compared to the methanol extracts (Figure 2B). Conversely, TAC was increased significantly (*p* ≤ 0.05) in the methanol extracts of *A. hypochondriacus*, *A. cruentus* and *A. spinosus* (Figure 2A).

The accumulation of TAC under combined drought and heat stress (at 30 °C, 35 °C and 40 °C) increased for *A. caudatus*, *A. hypochondriacus*, and *A. spinosus*. *A. cruentus* showed a significant decrease under similar conditions. In contrast, the highest increase was observed for *A. cruentus* under well-watered conditions (Figure 2A,C).

*A. caudatus*, *A. hypochondriacus* and *A. spinosus* showed higher sensitivity (at 35 °C and 40 °C), as the accumulation of TPC significantly (*p* ≤ 0.05) decrease with increasing temperature under well-watered conditions (Figure 3A).

The TPC accumulation increased significantly (*p* ≤ 0.05) with increasing temperature under combined drought and heat stress in *A. hypochondriacus*, *A. cruentus* and *A. spinosus*. In contrast, *A. caudatus* showed a significant (*p* ≤ 0.05) decrease under similar conditions (Figure 3C). TPC accumulation was substantially increased in the aqueous extracts of *A. spinosus* compared to methanol extracts (Figure 3A,B).

*A. cruentus* and *A. spinosus* showed less sensitivity to heat stress under well-watered conditions, as the accumulation of TFC increased significantly (*p* ≤ 0.05) with the increasing temperature in the methanol extracts. However, *A. hypochondriacus* showed higher sensitivity to heat stress, as the accumulation of TFC significantly (*p* ≤ 0.05) decreased with the increasing temperature under well-watered conditions (Figure 4A). Under similar conditions for both methanol and aqueous extracts, *A. caudatus* showed less sensitivity to heat stress (at 30 °C and 35 °C), but there was a significant (*p* ≤ 0.05) decrease of TFC accumulation with a further increase in temperature (at 40 °C) (Figure 4A–D).

The accumulation of TFC under combined drought and heat stress increased. *A. hypochondriacus* showed a significant (*p* ≤ 0.05) increase at 30 °C and 35 °C and a decrease at 40 °C temperature regimes. In contrast, *A. caudatus* showed a significant (*p* ≤ 0.05) decrease under similar conditions. TFC accumulation decreased (Figure 4C).

### 2.3. Influence of Drought and Heat Stress on Organic Acids and Flavonoids Compound Accumulation

In this study, an effort was made for the first time to compare the interactive effects of heat stress and combined drought and heat stress on the accumulation of 14 phenolic and flavonoid reference compounds at different temperature regimes on four African *Amaranthus* species. The values of phenolic and flavonoids compounds in *A. caudatus*, *A. hypochondriacus*, *A. cruentus* and *A. spinosus* were characterised by comparing the compounds with the masses of MRM Q1/Q3 ion pairs of these compounds’ reference standards. The absorption maxima of the UV-vis spectra of the reference compounds and the detected specific peaks of the corresponding components were also used to compare the identified compounds. A total of six phenolics (p-hydrobenzoic acid, caffeic acid, vanillic acid, p-coumaric acid, sinapic acid and ferulic acid) and five flavonoids (catechin, rutin, isoquercetin, quercetin and kampferol-3-O-rutinoside) compounds were identified. The individual phenolic and flavonoids compounds contents in different *Amaranthus* species under well-watered and drought-induced conditions treated with different temperature regimes were expressed as µg/g dry weight (DW) of plant material. Most compounds showed variation due to differences in *Amaranthus* species tested, treatments and solvents used for extraction. The most prominent observations were the major increase of caffeic acid detected only in *A. caudatus* and *A. hypochondriacus* (Table 1). The most predominant compound in all species was rutin, but a substantial amount of quercetin and ferulic acid were also identified in all tested species. A substantial amount of quercetin and ferulic acid were also identified in all tested species, followed by p-coumaric acid and isoquercetin that were only identified in *A. caudatus* and *A. hypochondriacus*. The phenolics p-hydrobenzoic acid, vanillic acid and sinapic acid, and flavonoids catechin and kampferol-3-O-rutinoside were significantly (*p* ≤ 0.05) lower in their accumulation when compared to other phenolic and flavonoids compounds (Table 1).

The accumulation of p-hydrobenzoic acid, vanillic acid, p-coumaric acid, sinapic acid and ferulic acid showed a significant (*p* ≤ 0.05) decrease with the increasing temperature in all tested species for both methanol and aqueous extracts under well-watered and drought-induced conditions compared to their respective controls. In contrast, accumulation of the flavonoids catechin, rutin, isoquercetin, quercetin and kampferol-3-O-rutinoside increased significantly (*p* ≤ 0.05) with the increasing temperature in all tested species for both methanol and aqueous extracts under similar conditions compared to their respective controls (Table 2, Table 3, Table 4 and Table 5). Caffeic acid accumulation could not be identified in methanol extracts of *A. hypochondriacus* under well-watered and drought-induced conditions. The aqueous extracts contained significant (*p* ≤ 0.05) amount of caffeic acid (248 µg/g) at 40 °C under well-watered conditions (Table 4). However, under combined drought and heat stress (Table 5) this compound significantly increased at 30 °C (532 µg/g) and decreased substantially at 35 °C (3.21 µg/g) and 40 °C (1.2 µg/g).

The extract yield of individual phenolics and flavonoids differ significantly (*p* ≤ 0.05) depending on the extraction solvent used across the tested *Amaranthus* species and various treatments. *A. caudatus* and *A. hypochondriacus* showed significant (*p* ≤ 0.05) increase in the accumulation of both phenolic and flavonoids compounds under well-watered and drought-induced conditions compared to *A. cruentus* and *A. spinosus* (Appendix A). Phenolic compounds p-hydrobenzoic acid, caffeic acid, vanillic acid, p-coumaric acid, sinapic acid and ferulic acid showed significant (*p* ≤ 0.05) accumulation in the aqueous extracts compared to the methanol extracts. The aqueous extracts from plants treated with combined drought and heat stress increased significantly (*p* ≤ 0.05) in the accumulation of these phenolic compounds. In contrast, flavonoids compounds rutin, isoquercetin, quercetin and kampferol-3-O-rutinoside increased significantly in the methanol extracts compared to the aqueous extracts. Catechin accumulation showed no significant differences between species, treatments and the solvent used for extraction (Appendix A).

## 3. Discussion

Vegetable *Amaranthus* contain significant levels of dietary minerals. The accumulation of mineral content was clearly influenced by various temperature regimes, as it decreased in parallel with increasing temperature. The *Amaranthus* species evaluated had higher values of K, Ca, Na and NO_3_- contents under control conditions and at 30 °C and 35 °C temperature regimes, but these values varied significantly (*p* ≤ 0.05) under all conditions (Figure 1A–D). The variation in K, Ca, Na and NO_3_- contents accumulation between *Amaranthus* species observed in this study was consistent with the results of the field-grown Choysum and Kailaan varieties [31], and *A. trycolor* genotype VA3 [4] exposed to drought stress.

In contrast, a significant (*p* ≤ 0.05) decline in the accumulation of K, Ca, and NO_3_− contents suggests sensitivity to heat stress (at 40 °C) and combined drought and heat stress (at 40 °C) (Figure 1A–D), although the decrease was more prominent for *A. caudatus* and *A. hypochondriacus*. Changes in the mineral content accumulation are associated with an increased electrolyte leakage (EL) [32,33]. This is an indication of membrane instability, which could be due to degradation in the lipid-protein configuration and loss of cellular function caused by decreased leaf relative water content (RWC) [34,35]. *A. cruentus* Ca content increased by 51.2% at 40 °C temperature regime (Figure 1C,G), and similar results were reported by [4] in *A. trycolor* genotype VA3.

Phenolic and flavonoid compounds are classified as powerful antioxidants due to their capability to scavenge free radicals, singlet oxygen, superoxide radicals and interaction with enzyme functions [36]. It is also well known that drought and heat stress and a combination of both trigger the accumulation of bioactive compounds, phenolics, flavonoids and antioxidant activity of many tolerant leafy vegetables [4,10,11,12,13,37]. However, drought and heat stress induce different behaviours in the concentrations and biological activity of these compounds depending on the species and their tolerance and/or sensitivity to these stressors [14,15,16,17,38]. The reducing capacity of a compound is usually an indicator of its potential antioxidant activity. Thus, high values obtained in this study for *A. caudatus*, *A. hypochondriacus* and *A. spinosus* suggests that they are potent sources of antioxidants. In addition, aqueous extraction showed to be better for TAC in *A. caudatus*. But TAC was better extracted with methanol in *A. hypochondriacus*, *A. cruentus* and *A. spinosus* (Figure 2A).

Heat stress encouraged the accumulation of TPC in *A. cruentus*. Conversely, under combined drought and heat stress *A. hypochondriacus*, *A. cruentus* and *A. spinosus* showed significant (*p* ≤ 0.05) increase in TPC accumulation, while *A. caudatus* showed a decrease as the temperature increases (Figure 3C). Aqueous extraction proved better for TPC in *A. spinosus*, and methanol extraction was better for *A. caudatus*, *A. hypochondriacus* and *A. cruentus* (Figure 3A,B).

TFC accumulation was substantial in both methanol and aqueous extracts of *A. cruentus* and *A. spinosus* under well-watered and drought-induced conditions. On the contrary, *A. caudatus* and *A. hypochondriacus* showed significant (*p* ≤ 0.05) decrease in the accumulation of TFC under similar conditions. These observations suggest that *A. cruentus* and *A. spinosus* are less sensitive to combined drought and heat stress than *A. caudatus* and *A. hypochondriacus*. In addition, methanol extraction yielded more TFC than aqueous extraction (Figure 4A–D).

In this study, it was observed that both drought and heat stress increased the accumulation of TAC, TPC and TFC, but it was dependent on the tolerance level of the species, the type of environmental stressor and severity of the stressor. Similar increases in TAC, TPC and TFC was reported in the *Achillea* species [18] and buckwheat [19] with an increase in drought stress. However, tolerant species performed better than the sensitive species.

Most of the identified phenolic and flavonoid compounds in this study were previously detected in leaves, stems, flowers, and seeds of *A. caudatus* [21,39,40], *A. hypochondriacus* [5,11,40], *A. cruentus* [20,41] and *A. spinosus* [42,43,44] grown under different conditions. In turn, various studies have identified gallic acid, coumarin and ellagic acid in leaves, stems, flowers, and seeds of *A. caudatus*, *A. hypochondriacus*, *A. cruentus* and *A. spinosus* [39,45,46]. In this study, gallic acid, coumarin and ellagic acid were below detection limits (0.089 µg/g DW) in all *Amaranthus* species tested. Similar results were reported by [21] in *A. caudatus* tested at seven stages of development. The results suggest that the accumulation of caffeic acid was dependent on the temperature regime under well-watered and/or drought-induced conditions. In turn, *A. hypochondriacus* was a better performing species than *A. caudatus* (Table 1, Table 2, Table 3, Table 4 and Table 5).

Rutin was the most predominant compound in all tested *Amaranthus* species, and this compound is considered one of the most potent antioxidants. Its levels are related to the developmental stages of the plant and increases as the plant ages [47,48,49]. The concentration of this compound differed significantly (*p* ≤ 0.05) between species and treatments. It increased in *A. caudatus*, *A. hypochondriacus* and *A. spinosus* and decreased in *A. cruentus* as the temperature increased under well-watered and drought-induced conditions. It is also worth noting that rutin was more prevalent in *A. caudatus* than all other species (Table 1). Further, the higher values observed in *Amaranthus* at a six leaf stage in this study under combined drought and heat stress were comparable with results previously reported in the older plants [5,21,47]. Various studies [5,21,47,48,49], including this study, have shown that *Amaranthus* leaves are an excellent source of rutin.

## 4. Materials and Methods

### 4.1. Chemicals and Reference Standards

Gallic acid (98%), DPPH (1,1-diphenyl-2-pycrylhydrazyl), tannic acid (98%), p-hydroxybenzoic acid (98%), ferulic acid (98%), vanillic acid (98%), rutin (98%), ellagic acid (98%), kaempferol-3-O-rutinoside (98%), quercetin (98%), isoquercetin (98%), (+)-catechin (98%), oleanic acid (98%), caffeic acid (98%), coumarin (98%), sinapic acid (98%), p-coumaric acid (98%), Folin–Ciocalteu’s phenol reagent, were purchased from Alfa Biotech (Chengdu, China), Industrial analytical (LGC group, Midrand, South Africa), and Sigma Aldrich (Baden-Württemberg, Germany). Aluminium chloride, sodium carbonate, sodium nitrite, sodium hydroxide and HPLC-grade solvents, including methanol, acetonitrile and formic acid were purchased from Associated Chemical Enterprises (Pty) Ltd. (Johannesburg, South Africa). and Merck (Pty) Ltd. (Darmstadt, Germany).

### 4.2. Plant Material and Growth Conditions

*Amaranthus* spp. (*A. caudatus, A. hypochondricus, A. cruentus* and *A. spinosus*) were seeded and maintained as previously described [33], and at the six-leaf stage after germination different temperature regimes ranging from 30 to 40 °C were introduced as the various heat stress treatment groups with plants at 26 °C as the control group. Although other C_4_ species are able tolerate temperatures up to 45 °C [50] 40 °C was used as the maximum temperature regime due to an irreversible damage of PSII structure observed at 40 °C under drought stress on *Amaranthus* species tested [33] Water was withheld for 7 days to induce drought stress (10% field capacity). Soil water content, conductivity and temperature were monitored every 60 min by ECH_2_O Data Logger (Model DEm50, Decagon Devices, Pullman, WA, USA) with soil moisture probes. Experiments were repeated three to five times (n), with three replicates for each species.

### 4.3. Determination of Mineral Content

Mineral content was measured using Horiba LAQUAtwin Meter (LAQUAtwin pH-22, Horiba Scientific, Tokyo, Japan). Samples were extracted by homogenising 0.02 g fresh leaf weight (FW) in 400 µL of sterilized ultrapure water. A sample volume of 300 µL was dispensed on the sensor reference to each mineral tested and covered by an adaptor on the top to measure the mineral content.

### 4.4. Determination of Total Phenolic Content, Total Flavonoid Content and Antioxidant Activity

#### 4.4.1. Sample Preparation

*Amaranthus* dried and crushed leaf samples (1.5 g) were combined with 30 mL of 80% [*v*/*v*] methanol [28,51,52]. Aqueous extract was prepared by combining 1.5 g of sample with 30 mL of water. The extraction was carried out on a Eumax Ultrasonic bath for 1 h at room temperature for both methanol and aqueous extracts. The mixtures were centrifuged at 4000 rpm for 10 min. The supernatants were filtered through a 0.45 µm PTFE membrane filter.

Methanol was evaporated using a rotary evaporator (BUCHI Rotavapor^®^ R-300, BÜCHI Labortechnik AG, 9230 Flawil, Switzerland). The methanol extract was concentrated using a SpeedVac concentrator (Thermo Scientific™ Savant™ SpeedVac Concentrator SPD121P 115, Waltham, MA, USA).

The collected filtrate of the aqueous extract was frozen overnight at −80 °C and the frozen extract was freeze-dried using a Virtis freeze dryer (SP Scientific, Gardiner, NY, USA). The dried extracts were kept at 4 °C and they were used as crude extracts for analyses of total phenolic content (TPC), total flavonoid content (TFC) and total antioxidant activity (TAC). All extractions were performed in triplicate using independent samples.

#### 4.4.2. Extraction of Flavonoids and Phenolic Compounds

Aqueous methanol extracts were prepared from all extracted samples. Briefly, ~0.1 g of lyophilized extract powder were dissolved with 5 mL of a methanol:water (50:50 *v*/*v*) mixture at room temperature for 30 min, while occasionally shaking. The extract was cooled for 10 min, and then centrifuged at 3000× *g* for 10 min. The supernatant was recovered, and the pellet was re-extracted for 45 min under the same conditions. Finally, the two supernatants were pooled and used for TAC, TPC, and TFC [53].

#### 4.4.3. Antioxidant Activity

The TAC of the methanol and aqueous extracts were estimated using the DPPH (1,1-diphenyl-2-pycrylhydrazyl) free radical-scavenging assay [54]. A quantity of 1 mg ascorbic acid standard was dissolved in 1 mL of double-distilled water (ddH_2_O). To a sample of 20 µL, 180 µL of DPPH (100 µmol/L) was added in a clear 96-well microplate. The mixture was shaken and allowed to stand at room temperature in the dark for 30 min. The absorbance of the resulting solution was measured at 517 nm against a blank (absolute methanol). The free radical-scavenging activity was calculated as follows:$$\mathrm{Scavenging}\;\mathrm{activity}\;(\%)=[1-(\frac{\mathrm{Absorbance}\;\mathrm{of}\;\mathrm{sample}}{\mathrm{Absorbance}\;\mathrm{of}\;\mathrm{control}})]\;\times \;100$$

A reference standard curve of ascorbic acid (vitamin C) was included (0.002 to 0.1 mg/mL), by plotting the percentage of free radical-scavenging activity of ascorbic acid versus its concentration. Results were expressed as mg vitamin C equivalent per 1 g dry weight (mg VCE/g DW).

### 4.5. Determination of Total Phenolic Content

TPC was determined using spectrophotometry [53]. A quantity of 1 mg gallic acid standard was dissolved in 1 mL of double-distilled water (ddH_2_O). To a sample of 10 µL, 47 µL of distilled water was added for a total volume of 57 µL. This was followed by the addition of 24 µL of Folin-Ciocalteu reagent (1 N) and 119 µL of sodium carbonate (20% Na_2_CO_3_) in a clear 96-well microplate. After 40 min at room temperature, the absorbance at 765 nm was read on a spectrophotometer against a blank that contained only methanol. A calibration curve was constructed within the concentration range of 0.025–0.25 mg/mL. The TPC values were expressed in milligrams of gallic acid equivalents/gram dry weight (mg GAE/g DW) of plant material using the equation:C = a×γ×(v/m),where C is the total amount of phenolic compounds (mg GAE/g DW sample), a is the dilution number, γ is the concentration obtained from the calibration curve (mg/mL), v is the volume of aqueous or methanol used for extraction (mL), and m is the weight of dry plant material (g).

### 4.6. Determination of Total Flavonoid Content

TFC was measured using a colorimetric assay [55]. A quantity of 1 mg catechin standard was dissolved in 1 mL ddH_2_O. To a sample of 11 µL, 45 µL distilled water was added to the clear 96-well microplate. At time 0 min, 3.5 µL of NaNO_2_ (5%) was added. After 5 min, 3.5 µL AlCl_3_ (10%) was added. At 6 min, 23 µL NaOH (1N) was added to the mixture. The solution was then diluted by adding 114 µL ddH_2_O and mixed thoroughly. The absorbance of the pink mixture was determined at 510 nm against a blank that contained distilled water. A calibration curve was constructed within the concentration range of 0.025–0.25 mg/mL. The TFC values for samples were expressed as milligrams of catechin equivalents/gram dry weight (mg CE/g DW) of plant material using the equation:C = a×γ×(v/m)where C is the total amount of flavonoid compounds (mg CE/g DW sample), a is the dilution number, γ is the concentration obtained from the calibration curve (mg/mL), v is the volume of aqueous or methanol used for extraction (mL), and m is the weight of dry plant material (g).

### 4.7. Sample Analysis by HPLC and LC-MS

100 mg of methanol and freeze-dried aqueous extracts were each separately dissolved in 1.5 mL of 99.9% methanol (HPLC grade). The samples were filtered with 0.45 µm syringe filters. The filtrate was used to analyse the phenolic and flavonoid compounds of the *Amaranthus* extracts using a HPLC-DAD Shimadzu system (Shimadzu, Kioto, Japan), consisting of a LC-2040 controller, DGU-403/405 degasser, LC-2040/C pump, LC-2040 autosampler, variable Shimadzu SPD-M30A diode array detector (DAD) and LC-2040 column oven. Phenolic and flavonoid reference compounds were injected (10 µL) into Venusil XBP C18 (2.1 × 100 mm, particle size 3 µm, Agilent, Santa Clara, CA, USA). The binary mobile phase consisted of solvents A (0.1% [*v*/*v*] of formic acid in water) and B (0.1% [*v*/*v*] of formic acid in acetonitrile) at a flow rate of 0.25 mL/min for a total run of 30 min. The system was run with the gradient program of 0–30 min starting with 10% B at time 0–4 min, 10–100% B at time 4–20 min, 100% B at time 20–25 min, 100–10% B at time 25–25.5 min and 10% B at time 25.5–30 min. The oven temperature was maintained at 40 °C and the detector wavelength ranged from 190–500 nm. Individual phenolic and flavonoid compound content in the leave extract was expressed on the basis of the calibration curve of the corresponding standards. The results were recorded as µg/g dry weight (DW).

Extracted compounds were identified through HPLC-MS/MS analysis that was carried out using a liquid chromatography (LC) system fitted with an Agilent 1260 Series HPLC-MS system (Agilent, Santa Clara, CA, USA) coupled with an Ultivo triple-quadrupole mass spectrometer (Ultivo LC/TQ LC-MS/MS system, Agilent, Santa Clara, CA, USA). The MS/MS operating conditions were set as follows: Nitrogen gas flow rate of 13 L/min at 350 °C, capillary voltage of 3000 V for both negative and positive ionisation, nebulizer pressure of 60 psi, and an ElectroSpray Ionization (ESI) source with polarity switching was used and a scan range of 100–1000 *m*/*z* was used.

### 4.8. Statistical Analysis

The data was analysed by using SigmaPlot version 12.0 (Systat Software, Inc., San Jose, CA, USA) software. The normality test (Shapiro-Wilk) one-way analysis of variance (ANOVA) was applied in all data in order to test the influence of drought and heat stress on mineral content and bioactive compounds composition at each temperature regime level. Comparison among means was determined through Fisher LSD method at *p* ≤ 0.05. The data in the figures and tables were expressed as mean ± standard deviation.

## 5. Conclusions

The response to environmental stressors of *A. caudatus*, *A. hypochondriacus*, *A. cruentus* and *A. spinosus* on the accumulation of mineral content, antioxidant activity, phenolic and flavonoid compounds varied according to the type and severity of the stress, solvent used for extraction and tolerance and/or sensitivity of genotype species tested. Efficient exploration of the tolerance level of *Amaranthus* to environmental stressors can provide economic and agricultural potential, high nutritional value and medicinal properties that can sustain communities. Combined drought and heat stress in *Amaranthus* species induced the accumulation of mineral content, antioxidant activity, phenolic and flavonoid compounds at 30 °C, 35 °C and 40 °C temperature regimes. *Amaranthus* species that showed sensitivity to heat stress at 40 °C showed a decrease in mineral content, antioxidant activity, phenolic and flavonoid accumulation. Results indicated that there is a direct relationship between total phenolic content and antioxidant activity. Heat stress at 40 °C under well-watered condition and combined drought and heat stress (at 30 °C) are the stress factors that induce the accumulation of caffeic acid. Heat stress increases the accumulation of rutin in *A. caudatus*, *A. hypochondriacus*, *A. cruentus* and *A. spinosus* under well-watered conditions and reduces under drought-induced conditions. These results confirm the previous claims that drought and heat stress and a combination of both trigger the accumulation of bioactive compounds. Methanol is a better solvent to extract flavonoid compounds, while an aqueous solution is a better extraction solvent for organic acids. This study demonstrated that *A. caudatus*, *A. hypochondriacus*, *A. cruentus* and *A. spinosus* can be used as vegetable source as they are a rich source of valuable minerals and bioactive compounds with beneficial effects on human health. *A. cruentus* and *A. spinosus* are more heat and drought tolerant than *A. caudatus* and *A. hypochondriacus*. Therefore, understanding the cultivar tolerance or sensitivity to drought and heat stress would be crucial in developing potential agricultural diversification with a better prospect of addressing food shortages and malnutrition.

## Figures and Tables

**Figure 1 plants-12-00953-f001:**
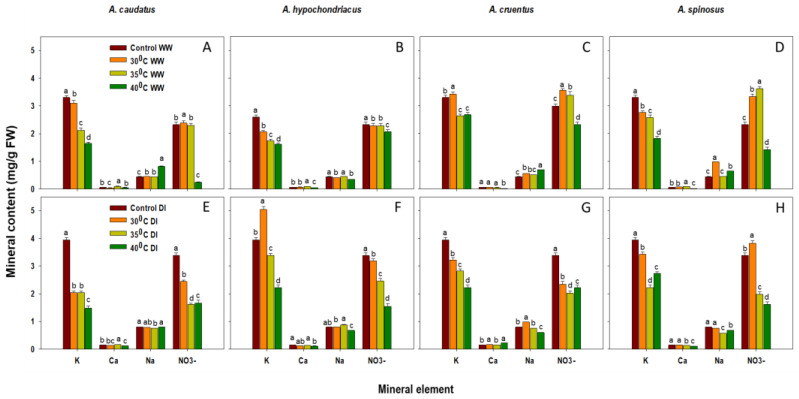
The effects of 30 °C, 35 °C and 40 °C temperature regimes under well-watered (WW) (**A**–**D**) and water-stressed (DI) (**E**–**H**) conditions on the mineral content of *A. caudatus*, *A. hypochondriacus*, *A. cruentus* and *A. spinosus*. Each vertical bar represents the mean value, and vertical line is a mean standard error (±) at 95% confidence level. Values among each species with the same letter(s) are not significantly different at *p* ≤ 0.05.

**Figure 2 plants-12-00953-f002:**
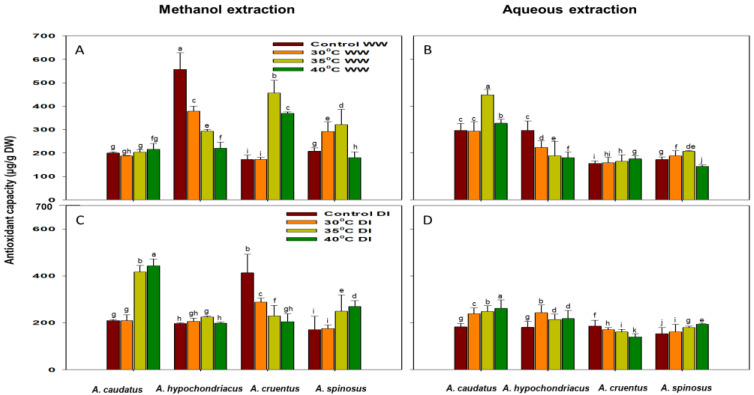
The effects of 30 °C, 35 °C and 40 °C temperature regimes under well-watered (WW) (**A**,**B**) and water-stressed (DI) (**C**,**D**) conditions on the accumulation of total antioxidant capacity (TAC) of *A. caudatus*, *A. hypochondriacus*, *A. cruentus* and *A. spinosus*. Each vertical bar represents the mean value, and vertical line is a mean standard error (±) at 95% confidence level. Values among each species with the same letter(s) are not significantly different at *p* ≤ 0.05.

**Figure 3 plants-12-00953-f003:**
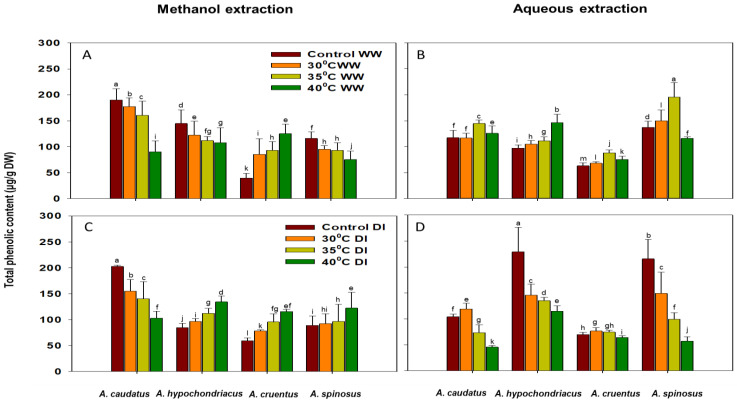
The effects of 30 °C, 35 °C and 40 °C temperature regimes under well-watered (WW) (**A**,**B**) and water-stressed (DI) (**C**,**D**) conditions on the accumulation of total phenolic content (TPC) of *A. caudatus*, *A. hypochondriacus*, *A. cruentus* and *A. spinosus*. Each vertical bar represents the mean value, and vertical line is a mean standard error (±) at 95% confidence level. Values among each species with the same letter(s) are not significantly different at *p* ≤ 0.05.

**Figure 4 plants-12-00953-f004:**
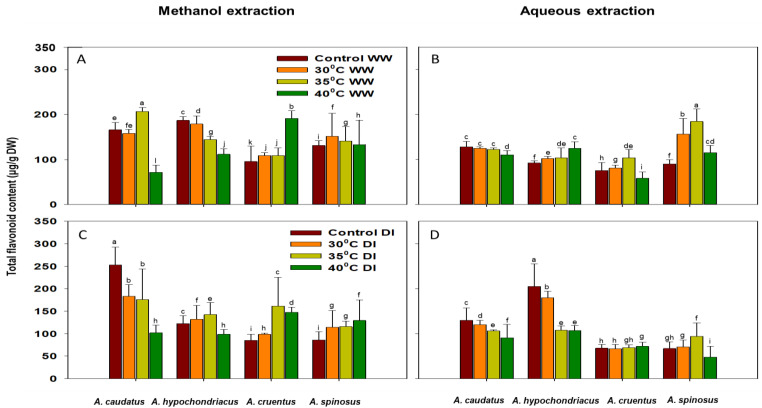
The effects of 30 °C, 35 °C and 40 °C temperature regimes under well-watered (WW) (**A**,**B**) and water-stressed (DI) (**C**,**D**) conditions on the accumulation of total flavonoid content (TFC) of *A. caudatus*, *A. hypochondriacus*, *A. cruentus* and *A. spinosus*. Each vertical bar represents the mean value, and vertical line is a mean standard error (±) at 95% confidence level. Values among each species with the same letter(s) are not significantly different at *p* ≤ 0.05.

**Table 1 plants-12-00953-t001:** Total phenolic and flavonoids compounds detected in leaves extract of *Amaranthus* species (µg/g DW). ND (not detected). Data are shown as mean ± standard deviation and different letters in the same row indicate significant differences (*p* ≤ 0.05).

Compound	*A. caudatus*	*A. hypochondriacus*	*A. cruentus*	*A. spinosus*
Catechin	28.80 ± 0.04 b	28.81 ± 0.03 bc	28.81 ± 0.02 c	28.80 ± 0.08 a
P-hydrobenzoic acid	33.38 ± 7.01 a	ND	ND	ND
Caffeic acid	109.91 ± 8.16 b	1318.80 ± 12.53 a	ND	ND
Vanillic acid	139.73 ± 24.32 a	14.16 ± 2.50 b	ND	ND
Rutin	550.61 ± 26.73 a	449.85 ± 24.30 b	363.29 ± 11.78 c	397.39 ± 78.42 b
P-coumaric acid	315.97 ± 40.28 a	57.75 ± 4.95 b	ND	ND
Isoquercetin	223.10 ± 17.15 a	106.20 ± 8.58 b	ND	ND
Quercetin	437.90 ± 26.31 a	412.30 ± 51.19 a	204.76 ± 33.67 b	88.11 ± 2.54 c
Kampferol-3-O-rutinoside	22.23 ± 2.66 d	47.30 ± 7.32 b	37.24 ± 6.15 c	65.08 ± 10.34 a
Sinapic acid	26.51 ± 5.90 a	ND	ND	ND
Ferulic acid	341.14 ± 30.30 a	154.20 ± 25.44 b	58.95 ± 9.39 c	35.14 ± 4.12 d
Gallic acid	ND	ND	ND	ND
Coumarin	ND	ND	ND	ND
Ellagic acid	ND	ND	ND	ND

**Table 2 plants-12-00953-t002:** Individual phenolic and flavonoids compounds contents detected MeOH leaves extracts of *Amaranthus* species treated with different temperature regimes under WW condition (µg/g DW). ND (not detected). Data are shown as mean ± standard deviation and different letters in the same row indicate significant differences (*p* ≤ 0.05).

Compound	*A. caudatus*				*A. hypochondriacus*				*A. cruentus*				*A. spinosus*			
	Control WW	30° WW	35° WW	40° WW	Control WW	30° WW	35° WW	40° WW	Control WW	30° WW	35° WW	40° WW	Control WW	30° WW	35° WW	40° WW
Catechin	1.8 ± 0.0040 a	1.8 ± 0.0014 b	1.8 ± 0.00061 b	1.8 ± 0.00040 b	1.8 ± 0.00024 b	1.8 ± 0.0040 a	1.8 ± 0.0033 a	1.8 ± 0.0041 a	1.8 ± 0.0026 a	1.8 ± 0.0023 a	1.8 ± 0.00023 b	1.8 ± 0.0030 a	1.8 ± 0.0069 a	1.8 ± 0.056 a	1.8 ± 0.0082 a	1.8 ± 0.0017 a
P-hydrobenzoic acid	14.3 ± 2.89 a	14.3 ± 2.85 a	2.0 ± 0.35 b	2.8 ± 0.92 b	ND	ND	ND	ND	ND	ND	ND	ND	ND	ND	ND	ND
Caffeic acid	11.3 ± 0.62 a	11.3 ± 0.68 a	6.6 ± 0.83 b	2.6 ± 0.75 c	ND	ND	ND	ND	ND	ND	ND	ND	ND	ND	ND	ND
Vanillic acid	12.3 ± 0.924 a	12.2 ± 0.902 a	8.1 ± 0.102 b	4.3 ± 1.13 c	ND	ND	ND	ND	ND	ND	ND	ND	ND	ND	ND	ND
Rutin	41.0 ± 2.04 f	38.5 ± 5.4 f	48.6 ± 1.4 d	51.0 ± 1.61 c	51.5 ± 0.29 cd	51.5 ± 0.27 cd	36.9 ± 0.83 g	120.0 ± 12.9 a	26.9 ± 0.094 h	26.8 ± 0.13 h	44.7 ± 0.058 f	44.7 ± 0.058 f	15.1 ± 0.96 j	15.1 ± 0.96 j	17.5 ± 0.25 i	92.1 ± 5.47 b
P-coumaric acid	9.8 ± 1.67 a	10.1 ± 1.99 a	8.3 ± 0.84 b	3.8 ± 1.81 c	ND	ND	ND	ND	ND	ND	ND	ND	ND	ND	ND	ND
Isoquercetin	13.1 ± 1.68 bd	13.2 ± 1.73 d	16.4 ± 1.72 c	19.4 ±2.01 b	12.7 ± 0.89 ae	12.7 ± 0.91 e	12.6 ± 0.35 f	68.2 ± 6.44 a	ND	ND	ND	ND	ND	ND	ND	ND
Quercetin	33.6 ± 1.49 d	33.7 ± 1.53 d	40.7 ± 2.11 c	43.5 ± 2.38 c	39.1 ± 8.03 b	39.2 ± 80 b	32.2 ± 1.44 e	114.0 ± 7.85 a	12.4 ± 1.18 h	12.5 ± 1.16 h	24.6 ± 3.34 f	19.9 ± 5.15 g	ND	ND	ND	ND
Kampferol-3-O-rutinoside	ND	ND	ND	ND	3.5 ± 0.71 de	3.7 ± 0.65 d	1.2 ± 1.41 g	14.4 ± 2.34 a	1.6 ± 0.49 f	1.8 ± 0.47 f	2.8 ± 0.96 e	8.49 ± 0.29 c	9.3 ± 4.09 b	9.1 ± 4.13 b	8.4 ± 0.60 c	9.39 ± 0.30 b
Sinapic acid	5.8 ± 10 a	5.5 ± 1.31 a	1.7 ± 0.43 b	2.26 ± 1.19 b	ND	ND	ND	ND	ND	ND	ND	ND	ND	ND	ND	ND
Ferulic acid	7.1 ± 1.98 ba	7.3 ± 1.80 a	5.0 ± 0.27 b	2.0 ± 0.92 c	ND	ND	ND	ND	ND	ND	ND	ND	ND	ND	ND	ND
Gallic acid	ND	ND	ND	ND	ND	ND	ND	ND	ND	ND	ND	ND	ND	ND	ND	ND
Coumarin	ND	ND	ND	ND	ND	ND	ND	ND	ND	ND	ND	ND	ND	ND	ND	ND
Ellagic acid	ND	ND	ND	ND	ND	ND	ND	ND	ND	ND	ND	ND	ND	ND	ND	ND

**Table 3 plants-12-00953-t003:** Individual phenolic and flavonoids compounds contents detected in MeOH leaves extracts of *Amaranthus* species treated with different temperature regimes under DI condition (µg/g DW). ND (not detected). Data are shown as mean ± standard deviation and different letters in the same row indicate significant differences (*p* ≤ 0.05).

Compound	*A. caudatus*				*A. hypochondriacus*				*A. cruentus*				*A. spinosus*			
	Control DI	30° DI	35° DI	40° DI	Control DI	30° DI	35° DI	40° DI	Control DI	30° DI	35° DI	40° DI	Control DI	30° DI	35° DI	40° DI
Catechin	1.8 ± 6.3 × 10^−6^ f	1.8 ± 0.012 c	1.8 ± 0.0071 b	1.8 ± 0.00887 b	1.8 ± 0.00051 d	1.8 ± 0.00051 d	1.8 ± 0.0024 b	1.8 ± 0.001 c	1.8 ± 0.001 c	1.81 ± 0.0014 a	1.8 ± 0.0068 b	1.8 ± 0.0015 c	1.8 ± 0.00022 e	1.8 ± 0.0015 c	1.8 ± 0.00047 d	1.8 ± 0.00057 de
P-hydrobenzoic acid	ND	ND	ND	ND	ND	ND	ND	ND	ND	ND	ND	ND	ND	ND	ND	ND
Caffeic acid	4.25 ± 0 c	4.28 ± 0.043 bc	7.05 ± 0.12 a	4.54 ± 0.27 b	ND	ND	ND	ND	ND	ND	ND	ND	ND	ND	ND	ND
Vanillic acid	1.96 ± 0.84 c	3.04 ± 0.22 b	3.61 ± 0.053 b	9.16 ± 4.11 a	ND	ND	ND	ND	ND	ND	ND	ND	ND	ND	ND	ND
Rutin	79.5 ± 4.64 c	79.7 ± 4.6 c	106 ± 5.33 a	87.9 ± 1.11 c	73.9 ± 2.57 d	74.2 ± 2.13 d	73.3 ± 1.3 gd	46 ± 3.26 g	51.6 ± 4.63 f	51.6 ± 4.56 f	67.2 ± 0.53 e	35.6 ± 0.90 i	71.1 ± 28.4 b	71.2 ± 28.40 b	40.2 ± 1.14 h	33 ± 3.06 i
P-coumaric acid	1.66 ± 1.41 b	1.7 ± 1.38 b	3.01 ± 2.52 a	1.75 ± 0.058 c	ND	ND	ND	ND	ND	ND	ND	ND	ND	ND	ND	ND
Isoquercetin	35.8 ± 3.21 b	35.8 ± 3.24 b	49.8 ± 3.42 a	39.6 ± 0.14 b	ND	ND	ND	ND	ND	ND	ND	ND	ND	ND	ND	ND
Quercetin	64 ± 5.30 b	64 ± 5.32 b	87.9 ± 7.77 a	70.5 ± 0.41 b	51.5 ± 9.93 c	51.5 ± 9.88 c	56.1 ± 2.01 c	28.7 ± 4.05 f	32.8 ± 6.64 e	32.9 ± 6.67 e	39.3 ± 7.88 d	16.4 ± 0.218 g	5.79 ± 0.68 i	5.82 ± 0.68 i	12.2 ± 0.073 h	12.4 ± 0.20 h
Kampferol-3-O-rutinoside	5.02 ± 0.68 e	5.27 ± 1.01 e	9.49 ± 0.67 b	2.45 ± 0.31 g	7.36 ± 0.42 c	7.61 ± 0.18 c	8.09 ± 0.74 b	1.53 ± 0.89 h	2.26 ± 1.69 fg	2.51 ± 2.17 ef	12 ± 0.0056 a	5.95 ± 0.073 d	2.6 ± 0.23 f	2.85 ± 0.22 f	5.12 ± 0.17 e	1.97 ± 0.25 h
Sinapic acid	3.58 ± 0.018 a	3.83 ± 0.49 a	1.28 ± 0.39 c	2.62 ± 1.07 b	ND	ND	ND	ND	ND	ND	ND	ND	ND	ND	ND	ND
Ferulic acid	6.95 ± 1.14 a	7.45 ± 0.39 a	6.22 ± 2.59 b	4.41 ± 1.14 d	ND	ND	ND	ND	2.5 ± 0.26 e	2.52 ± 0.26 e	5.92 ± 0.35 c	5.36 ± 0.32 cd	ND	ND	ND	ND
Gallic acid	ND	ND	ND	ND	ND	ND	ND	ND	ND	ND	ND	ND	ND	ND	ND	ND
Coumarin	ND	ND	ND	ND	ND	ND	ND	ND	ND	ND	ND	ND	ND	ND	ND	ND
Ellagic acid	ND	ND	ND	ND	ND	ND	ND	ND	ND	ND	ND	ND	ND	ND	ND	ND

**Table 4 plants-12-00953-t004:** Individual phenolic and flavonoids compounds contents detected in aqueous leaves extracts of *Amaranthus* species treated with different temperature regimes under WW condition (µg/g DW). ND (not detected). Data are shown as mean ± standard deviation and different letters in the same row indicate significant differences (*p* ≤ 0.05).

Compound	*A. caudatus*				*A. hypochondriacus*				*A. cruentus*				*A. spinosus*			
	Control WW	30° WW	35° WW	40° WW	Control WW	30° WW	35° WW	40° WW	Control WW	30° WW	35° WW	40° WW	Control WW	30° WW	35° WW	40° WW
Catechin	1.8 ± 0.0012 a	1.8 ± 0.0012 a	1.8 ± 9.2 × 10^−6^ e	1.8 ± 0.00042 b	1.8 ± 0.000012 d	1.8 ± 0.00012 c	1.8 ± 0.00020 b	1.8 ± 0.0022 a	1.8 ± 0.00025 b	1.8 ± 0.000091 cd	1.8 ± 0.00017 b	1.8 ± 0.000067 bc	1.8 ± 0.00025 a	1.8 ± 0.0016 b	1.8 ± 0.000086 cd	1.8 ± 0.00011 c
P-hydrobenzoic acid	ND	ND	ND	ND	ND	ND	ND	ND	ND	ND	ND	ND	ND	ND	ND	ND
Caffeic acid	7.68 ± 1.39 e	7.71 ± 1.43 e	6.26 ± 0.0093 d	5.87 ± 1.38 c	1.39 ± 0.82 a	1.4 ± 0.82 a	3.6 ± 10 b	248 ± 1.68 f	ND	ND	ND	ND	ND	ND	ND	ND
Vanillic acid	9.16 ± 4.11 b	9.1 ± 4.11 b	18.2 ± 6.84 a	11.6 ± 0.57 b	1.56 ± 0.12 e	1.51 ± 0.11 f	1.98 ± 0.52 d	3.02 ± 0.50 c	ND	ND	ND	ND	ND	ND	ND	ND
Rutin	3 ± 0.022 bc	3.25 ± 0.49 b	1.75 ± 0.018 f	2.18 ± 0.046 d	2.02 ± 0.081 d	2.07 ± 0.13 d	1.6 ± 0.020 g	2.38 ± 0.037 c	1.84 ± 0.24 e	1.87 ± 0.19 e	1.8 ± 0.15 ef	1.86 ± 0.14 e	11.5 ± 0.014 a	12 ± 0.015 a	1.84 ± 0.13 e	1.67 ± 0.14 f
P-coumaric acid	34.5 ± 2.55 b	34.6 ± 2.57 b	43.9 ± 9.74 a	44 ± 7.36 a	13.9 ± 0.082 c	13.9 ± 0.14 c	3.76 ± 0.49 d	1.73 ± 1.28 d	ND	ND	ND	ND	ND	ND	ND	ND
Isoquercetin	ND	ND	ND	ND	ND	ND	ND	ND	ND	ND	ND	ND	ND	ND	ND	ND
Quercetin	ND	ND	ND	ND	ND	ND	ND	ND	3.1 ± 0 d	3.12 ± 0.02 d	3.62 ± 0.59 c	4.12 ± 0.82 b	13 ± 0.15 a	12.9 ± 0.19 a	13 ± 0.13 a	13 ± 0.45 a
Kampferol-3-O-rutinoside	ND	ND	ND	ND	ND	ND	ND	ND	ND	ND	ND	ND	6.93 ± 0.15 a	6.91 ± 0.19 a	1.128 ± 0.0021 b	1.448 ± 0.0021 b
Sinapic acid	ND	ND	ND	ND	ND	ND	ND	ND	ND	ND	ND	ND	ND	ND	ND	ND
Ferulic acid	50.5 ± 0.21 a	48 ± 4.9 a	45 ± 2.65 a	30 ± 6.05 c	20.3 ± 8.5 d	21.6 ± 7.63 d	15.9 ± 0.47 e	14.3 ± 2.90 e	12.1 ± 2.35 f	12.2 ± 2.30 f	11.5 ± 2.33 g	6.85 ± 1.22 h	4.71 ± 0.43 j	4.96 ± 0.80 j	5.33 ± 0.34 i	4.55 ± 0.30 j
Gallic acid	ND	ND	ND	ND	ND	ND	ND	ND	ND	ND	ND	ND	ND	ND	ND	ND
Coumarin	ND	ND	ND	ND	ND	ND	ND	ND	ND	ND	ND	ND	ND	ND	ND	ND
Ellagic acid	ND	ND	ND	ND	ND	ND	ND	ND	ND	ND	ND	ND	ND	ND	ND	ND

**Table 5 plants-12-00953-t005:** Individual phenolic and flavonoids compounds contents detected in aqueous leaves extracts of *Amaranthus* species treated with different temperature regimes under DI condition (µg/g DW). ND (not detected). Data are shown as mean ± standard deviation and different letters in the same row indicate significant differences (*p* ≤ 0.05).

Compound	*A. caudatus*				*A. hypochondriacus*				*A. cruentus*				*A. spinosus*			
	Control DI	30° DI	35° DI	40° DI	Control DI	30° DI	35° DI	40° DI	Control DI	30° DI	35° DI	40° DI	Control DI	30° DI	35° DI	40° DI
Catechin	1.8 ± 0.0011 c	1.8 ± 0.0021 b	1.8 ± 0.00029 e	1.8 ± 0.00029 e	1.8 ± 0.00035 e	1.8 ± 0.00012 f	1.8 ± 0.0011 c	1.81 ± 0.0078 a	1.8 ± 0.0014 bc	1.8 ± 0.0015 bc	1.8 ± 0.00046 de	1.8 ± 0.00047 de	1.8 ± 0.00068 d	1.8 ± 0.00077 d	1.8 ± 0.00011 f	1.8 ± 0.00004 g
P-hydrobenzoic acid	ND	ND	ND	ND	ND	ND	ND	ND	ND	ND	ND	ND	ND	ND	ND	ND
Caffeic acid	10.7 ± 0.058 c	10.8 ± 0.058 c	4.7 ± 0.53 d	4.25 ± 0 e	528 ± 3.63 b	532 ± 3.66 a	3.21 ± 0.18 f	1.2 ± 0.74 g	ND	ND	ND	ND	ND	ND	ND	ND
Vanillic acid	13.5 ± 0.091 a	13.5 ± 0.082 a	2.82 ± 0.23 c	7.19 ± 0 b	1.34 ± 0.099 f	1.34 ± 0.098 f	1.23 ± 0.58 e	2.18 ± 0.49 d	ND	ND	ND	ND	ND	ND	ND	ND
Rutin	2.23 ± 0.0019 d	2.23 ± 0.0067 d	1.73 ± 0.018 bf	2.04 ± 0 e	4.78 ± 0.22 b	4.8 ± 0.19 b	2.43 ± 0.0086 c	2.47 ± 0.051 c	1.71 ± 0.03 g	1.75 ± 0.063 f	1.71 ± 0.010 g	1.65 ± 0.013 gh	1.52 ± 0.021 h	1.53 ± 0.024 h	7.29 ± 0.40 a	4.74 ± 0.44 b
P-coumaric acid	54.9 ± 2.92 a	55 ± 2.91 a	4.3 ± 0.55 d	4.67 ± 0 d	5.6 ± 0.50 cd	5.61 ± 0.51 c	6.33 ± 1.94 b	6.92 ± 0.0089 bc	ND	ND	ND	ND	ND	ND	ND	ND
Isoquercetin	ND	ND	ND	ND	ND	ND	ND	ND	ND	ND	ND	ND	ND	ND	ND	ND
Quercetin	ND	ND	ND	ND	ND	ND	ND	ND	ND	ND	ND	ND	ND	ND	ND	ND
Kampferol-3-O-rutinoside	ND	ND	ND	ND	ND	ND	ND	ND	ND	ND	ND	ND	ND	ND	ND	ND
Sinapic acid	ND	ND	ND	ND	ND	ND	ND	ND	ND	ND	ND	ND	ND	ND	ND	ND
Ferulic acid	53.1 ± 1.32 b	55.6 ± 4.53 a	9.72 ± 0.41 e	2.89 ± 0 g	24 ± 1.35 c	24.6 ± 1.26 c	16.5 ± 1.15 d	17 ± 2.18 d	ND	ND	ND	ND	5.07 ± 0.66 f	5.32 ± 0.83 f	4.34 ± 0.68 f	0.861 ± 0.078 h
Gallic acid	ND	ND	ND	ND	ND	ND	ND	ND	ND	ND	ND	ND	ND	ND	ND	ND
Coumarin	ND	ND	ND	ND	ND	ND	ND	ND	ND	ND	ND	ND	ND	ND	ND	ND
Ellagic acid	ND	ND	ND	ND	ND	ND	ND	ND	ND	ND	ND	ND	ND	ND	ND	ND

## Data Availability

The online version contains supplementary material available at URL: https://repository.nwu.ac.za/.

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
