# Peer review of "Influence of Drought and Heat Stress on Mineral Content, Antioxidant Activity and Bioactive Compound Accumulation in Four African *Amaranthus* Species"

_plants, 2023, doi:10.3390/plants12040953_

Round 1

Reviewer 1 Report

first of all i want to thank all the authors for this interesting paper

Any comment will just to improve the overall quality

1- Abstract is poor  and the gap of knowladge is not clear the value of the heat and drought stress is absent

2- Introduction need to be enhanced and stating why u want to study the combined effect and have to state the global warming and the circustancesse that evolved and problem that we face now day.

3- Also you have to state why u chosse such crop and it economic value and it face such problems.

3- Material and method is very poor regarding the experimental desigin - i dont know hoe u made the stress (why u chosse these temperatures ) clearly and all the conditon of the palnt growth.

4- Result is very hard to follow to separate well watered from the drought stress and those combined make it vey hard to detect the combined effect. I suggest to change the figures to put the well water and the drought in the same figure fro each paprameter to be able to compare and follow the combined effect.

Disscusion is ok, although it look discriptive and no action mechanisme for what happend under the combined stress in each spieces. need to draw a summary figure to disscuse the mechanism.

Reviewer 2 Report

Dear Authors,

 After go through the assigned manuscript “Influence of drought and heat stress on mineral content, antioxidant activity and bioactive compound accumulation in four African Amaranthus species” I feel that research out comes is very interesting and citable in the future. Although, there are many weak points throughout the manuscript, those required proper attention and careful revision before the final publication.

Observations,

In the title as well as in whole ms, instead of term “drought” moisture stress is more appropriate term since drought is very complex and meteorological term.

Abstract is reflecting only a part of the result. It should not be the only results of your experiment, It should be summary of research (problem, hypothesis, experiment, result, application and future perspective).

Introduction, In this section role of studied phenolic and flavonoid compounds in human health and plant stress tolerance must be placed with special emphasis to caffeic acid and rutin. The objective and hypothesis of the research work is also lacking.

The methodology should contain information regarding experimental design to validate the hypothesis, treatments imposed (moisture and temperature stress). Where the experiments were conducted (in open or in controlled condition) also define the position of the experiment i.e. in field or pots (size of pots)? Further no details are mentioned on how treatments effects were compared??  How minimum degree of freedom was maintained for test of significance in ANOVA?? How soil water potential was maintained in pots or in field? All the treatments must be clearly defined. These general questions need to be answered.

How the samples were prepared for mineral content estimation, procedure must be defined with proper reference for each mineral.

Whether it is feasible that total phenolic content and total flavonoid content can be estimated through same units, kindly justify. In the section determination of total phenolic content, statement “Where C is the total amount of flavonoid compounds (mg GAE/g DW sample)” same as in the section determination of total flavonoid content, statement “Where C is the total amount of flavonoid compounds (mg GAE/g DW sample)” kindly clarify

Statistical analysis: I cannot understand what the authors are trying to communicate in the statement “The normality test (Shapiro-Wilk) one-way analysis of variance (ANOVA) was applied in all data to test the influence of drought and heat stress on mineral content and bioactive compounds composition at each temperature regime level. Does the authors clarify that what it means “The normality test (Shapiro-Wilk) one-way analysis of variance (ANOVA)”.

Results section: authors stated that “ All species had higher values of K, Ca, Na and NO3- contents at 30°C and 35°C, but these values varied significantly (p ≤ 0.05) under almost all conditions. Both species showed higher sensitivity to heat stress (at 40°C), as the accumulation of K, Ca and NO3- decreased under well-watered conditions compared to their respective controls” I cannot understand what the authors are trying to communicate here.  Please define the respective controls of well-watered conditions.

Figures: In the entire figures, reverse order of the alphabet would be more appropriate. Means “a” would be on the top

Tables: In the title of table 1, “Total phenolic and flavonoids compounds detected in leaves extract of Amaranthus species under well-watered and drought-induced conditions” in mentioned, however after seeing the table it is not clear that the values are obverse under  well-watered condition or in drought-induced conditions, kindly clarify. Instead of inverted comma, decimal must be there for all the values.

The values of Catechin content are same in all the species, whether it will be feasible. Please also mentioned the name of the statistical test through which content of compounds was compared. 

 In the tables 2, 3, and 4 it can also be seen that the values of catechin content are same in all the treatments, could it be possible .Please clarify.

Conclusion section: The conclusions section should not be a summary of your study. conclusion is not providing any take home message, means how the researcher  can utilized the key findings in future studies. The conclusions section should illustrate the mechanistic links of findings obtained under applied treatments.

Round 2

Reviewer 1 Report

The paper is much better

congratulation

Reviewer 2 Report

The normality test and one way ANOVA both are different test, therefore both test must be separated in the methodology section.

Except this all the comments have been addressed properly. I have no further comments